# 13-(2-Methylbenzyl) Berberine Is a More Potent Inhibitor of MexXY-Dependent Aminoglycoside Resistance than Berberine

**DOI:** 10.3390/antibiotics8040212

**Published:** 2019-11-06

**Authors:** Kenta Kotani, Mio Matsumura, Yuji Morita, Junko Tomida, Ryo Kutsuna, Kunihiko Nishino, Shuji Yasuike, Yoshiaki Kawamura

**Affiliations:** 1Department of Microbiology, School of Pharmacy, Aichi Gakuin University, Nagoya 560-0043, Japan; ag163a01@dpc.agu.ac.jp (K.K.); jtomida@dpc.agu.ac.jp (J.T.); kutsuna@dpc.agu.ac.jp (R.K.); 2Department of Organic and Medicinal Chemistry, School of Pharmacy, Aichi Gakuin University, Nagoya 560-0043, Japan; m-matsu@dpc.agu.ac.jp (M.M.); s-yasuik@dpc.agu.ac.jp (S.Y.); 3Department of Microbial Science and Host Defense, Meiji Pharmaceutical University, Noshio, Kiyose, Tokyo 560-0043, Japan; morita@my-pharm.ac.jp; 4Department of Biomolecular Science and Regulation, Institute of Scientific and Industrial Research, Osaka University, Osaka 560-0043, Japan; nishino@sanken.osaka-u.ac.jp

**Keywords:** *Pseudomonas aeruginosa*, efflux, MexXY, aminoglycoside resistance, berberine

## Abstract

We previously showed that berberine attenuates MexXY efflux-dependent aminoglycoside resistance in *Pseudomonas aeruginosa*. Here, we aimed to synthesize berberine derivatives with higher MexXY inhibitory activities. We synthesized 11 berberine derivatives, of which 13-(2-methylbenzyl) berberine (13-o-MBB) but not its regiomers showed the most promising MexXY inhibitory activity. 13-o-MBB reduced the minimum inhibitory concentrations (MICs) of various aminoglycosides 4- to 128 fold for a highly multidrug resistant *P. aeruginosa* strain. Moreover, 13-o-MBB significantly reduced the MICs of gentamicin and amikacin in *Achromobacter xylosoxidans* and *Burkholderia cepacia*. The fractional inhibitory concentration indices indicated that 13-o-MBB acted synergistically with aminoglycosides in only MexXY-positive *P. aeruginosa* strains. Time-kill curves showed that 13-o-MBB or higher concentrations of berberine increased the bactericidal activity of gentamicin by inhibiting MexXY in *P. aeruginosa*. Our findings indicate that 13-o-MBB inhibits MexXY-dependent aminoglycoside drug resistance more strongly than berberine and that 13-o-MBB is a useful inhibitor of aminoglycoside drug resistance due to MexXY.

## 1. Introduction

*Pseudomonas aeruginosa* is a major cause of nosocomial infections. Treatment of *P. aeruginosa* infections with antimicrobial concentrations insufficient to inhibit *P. aeruginosa* growth results in the emergence of new multidrug resistant *P. aeruginosa* strains [1] that are difficult to eradicate and may increase mortality [2].

Drug efflux is a major mechanism leading to antimicrobial resistance in *P. aeruginosa* [3]. Four resistance-nodulation-division (RND)-type multidrug efflux pumps (MexAB-OprM [4], MexCD-OprJ [5], MexEF-OprN [6] and MexXY-OprM/OprA [7,8]) have been reported as drug efflux systems involved in the drug resistance of *P. aeruginosa*. Of these, only MexXY contributes to aminoglycoside drug resistance [8,9]. The MexXY-OprM system comprises a cytoplasmic membrane antibiotic-proton antiporter (MexY), an outer membrane porin (OprM), and a periplasmic membrane fusion protein (MexX) [10]. MexXY has multiple functions, including the expulsion of antibiotics. Wild-type *P. aeruginosa* expresses low MexXY levels but elevated MexXY has been detected in aminoglycoside-resistant *P. aeruginosa* strains [11,12]. Therefore, the development of MexXY inhibitors would allow the use of lower concentrations of aminoglycoside drugs that can cause severe side effects such as kidney damage [13].

There have been various reports of inhibitors of RND-type multidrug efflux pumps, but no clinical applications have been published to date [14]. Phenyl-arginine-β-naphthylamide (PAβN, MC-207,110), a well-known efflux pump inhibitor, does not inhibit aminoglycoside resistance due to MexXY [15]. We previously reported that berberine attenuates MexXY-dependent aminoglycoside resistance in *P. aeruginosa* [15], consistent with a recent report that berberine has high affinity to a MexXY model protein in silico [16].

Berberine is an isoquinoline quaternary alkaloid isolated from many kinds of medicinal plants such as *Coptis chinensis*, *Coptis rhizome*, *Coptis japonica* and *Phellondendron amurense* [17] and has weak antibacterial activity against Gram-negative bacteria such as *P. aeruginosa* [18]. Various derivatives of berberine have been developed and studied for their anti-hyperglycemic, anti-cancer, anti-inflammatory, anti-Alzheimer’s disease and anti-microbial activities [19]. Derivatives with multidrug resistance pump inhibitory activity against *Staphylococcus aureus* [20] and that reduce fluconazole resistance against *Candida albicans* [21] have been reported. In addition, quaternary ammonium compounds inhibit the biofilm formation in *P. aeruginosa* and *C. albicans* have been reported [22].

The optimum concentration of berberine to inhibit MexXY in *P. aeruginosa* cells is more than 512 µg/mL [15], which is too high for clinical application. In this study, we aimed to synthesize berberine derivatives with higher MexXY inhibitory activities.

## 2. Results

### 2.1. Antibacterial Activity of Berberine Derivatives toward P. aeruginosa

We first measured the minimum inhibitory concentrations (MICs) of 11 berberine derivatives (Figure 1) synthesized against *P. aeruginosa* mutants PAGU^g^1927, which expresses MexXY, and PAGU^g^1931, which does not express MexXY. A difference in the activity of a derivative toward the two strains indicates that the MexXY activity is not masked by the other four pumps (MexAB, MexCD, MexEF and MexVW) [15]. The MIC values of the berberine derivatives were lower in both strains compared to berberine (Table 1), suggesting that these berberine derivatives had higher anti-pseudomonas activity compared with berberine. These berberine derivatives showed similar MIC values that differed no more than, 4-fold. Their MIC values against PAGU^g^1927 were 2-fold greater than against PAGU^g^1931, indicating that the derivatives are MexXY substrates.

### 2.2. Inhibition of Drug Resistance in P. aeruginosa Using Combined Berberine Derivatives

We investigated the MexXY inhibitory activities of the berberine derivatives by measuring the MICs of gentamicin in the presence of the derivatives against *P. aeruginosa* mutants PAGU^g^1927 and PAGU^g^1931 (Table 2). The concentrations of the berberine derivatives were 1/2, 1/4, or 1/8 that of the MICs for PAGU^g^1931.

The MIC of gentamicin for PAGU^g^1927 in the presence of 256 µg/mL berberine was 128 µg/mL (Table 2), which is one-eighth that of gentamicin alone (Table 1). Compounds 1–5 and 7 exhibited apparently increased MexXY inhibitory activity, with compound 7 reducing the MIC of gentamicin 64-fold. Compound 7 was named 13-o-MBB. Compounds 8 and 9, which are regioisomers of 13-o-MBB, increased sensitivity to gentamicin by up to 4-fold and were the weakest MexXY inhibitors.

We also examined changes in sensitivity to drugs other than gentamicin by combination with the berberine derivatives (Table 3). The combined use of 13-o-MBB 128 µg/mL reduced the MIC values of various substrate drugs (amikacin, tobramycin, kanamycin, gentamicin, spectinomycin, norfloxacin, ciprofloxacin, erythromycin, carbenicillin, ethidium bromide, tetracycline, chloramphenicol, azithromycin and cefepime) targeting MexXY by 2-fold to 16-fold (Table 3). The regiosomer 13-(3-methylbenzyl) berberine bromide (13-m-MBB) increased the spectinomycin sensitivity of PAGU^g^1927 8-fold and that of cefepime 4-fold at 64 µg/mL, whereas the other derivatives did not change the sensitivity to spectinomycin more than 2-fold. In addition, the combined use of 64 µg/mL of 13-(4-methyl-benzyl)-berberine bromide (13-p-MBB) increased sensitivity t cefepime 4-fold for PAGU^g^1927. Moreover, the sensitizing action of 13-(3-methyl-benzyl)-berberine bromide and 13-(4-methyl-benzyl)-berberine bromide did not exceed that of 13-o-MBB for PAGU^g^1927. Taken together, these results suggest that the *o*-methyl group of 13-o-MBB increases antimicrobial sensitivity in a MexXY-dependent manner.

We investigated whether the inhibitory action of 13-o-MBB against MexXY-dependent drug resistance can be observed in PAGU 1606, a multidrug resistant *P. aeruginosa* clinical strain, and its MexXY-deficient strain PAGU^g^1659 (Table 4). The MIC of amikacin alone against PAGU 1606 was 256 µg/mL and 64 µg/mL when combined with berberine. In contrast, the combined use of 13-o-MBB and amikacin decreased the MIC to 16 µg/mL. Thus, 13-o-MBB inhibits amikacin resistance 4-fold more effectively than berberine in a MexXY-dependent drug resistant strain. Another aminoglycoside drug, 13-o-MBB, inhibited drug resistance two to four times stronger than berberine but had no greater effect on the drug resistance of PAGU 1606 than the other aminoglycosides. However, the MICs of norfloxacin, erythromycin and azithromycin were increased towards PAGU^g^1659, a pump-deficient strain.

The sensitizing action of 13-o-MBB for various aminoglycosides was compared with that of berberine at the same concentrations as tested against *P. aeruginosa* clinical strains but using *Burkholderia cepacia* PAGU 0013 and *Achromobacter xylosoxidans* PAGU 0002 (Table 5). The two non-*P. aeruginosa* strains are naturally resistant to aminoglycosides due to the presence of MexXY orthologs [7,23]. 13-o-MBB at the same concentration as berberine increased the sensitivity to the aminoglycosides more than 4-fold over that of berberine. In addition, comparison of the MICs of the aminoglycosides in combination with 13-o-MBB towards a clinical strain of *P. aeruginosa* and its *mexXY*-deficient strain provided similar MIC values. 13-o-MBB greatly increased the sensitivity to aminoglycoside drugs for *P. aeruginosa, B. cepacia*, and *A. xylosoxidans*, increasing the sensitivity to amikacin more than 128-fold and to gentamycin more than 512-fold for *A. xylosoxidans*.

### 2.3. Interaction between 13-o-MBB and Aminoglycoside Drugs

The fractional inhibitory concentration (FIC) values were determined using 13-o-MBB or berberine and gentamicin or amikacin in combination with *P. aeruginosa* strains PAGU 1606 and PAGU^g^1927 and their MexXY-defective mutants PAGU^g^1659 and PAGU^g^ 1931 (Table 6). The combination of 13-o-MBB and amikacin or gentamicin showed a synergistic effect in the MexXY-expressing strain, showing that the MexXY-dependent aminoglycoside resistance inhibitory action of 13-o-MBB is synergistic. In addition, the MICs of 13-o-MBB and berberine were reduced only in combination with amikacin or gentamicin and only in the MexXY-expressing strain, showing that the combination of amikacin or gentamicin in the presence of MexXY increases the accumulation of 13-o-MBB and berberine in the cell.

### 2.4. Time-Killing Assay

The bactericidal activity of gentamicin together with berberine and 13-o-MBB against *P. aeruginosa* was investigated using PAGU^g^1933 and PAGU^g^1929. PAGU^g^1933 was killed after 4 h treatment with gentamicin at 2 µg/mL whereas the growth of PAGU^g^1929 was suppressed but no bactericidal action was observed (Figure 2). Treatment of PAGU^g^1929 for 4 h with 2 µg/mL gentamicin in combination with 256 µg/mL berberine reduced the number of colonies about 100-fold. In addition, treatment of PAGU^g^1929 with a combination of 2 µg/mL gentamicin and 64 µg/mL 13-o-MBB enhanced the bactericidal action of gentamicin more than 10-fold over that of 256 µg/mL berberine.

## 3. Discussion

The addition of 128 µg/mL 13-o-MBB increased the sensitivity to aminoglycosides by 2-fold to 8-fold in comparison with 256 µg/mL berberine in the MexXY-positive *P. aeruginosa* strain PAGU^g^1927 (Table 2 and Table 3). The antimicrobial activity of 13-o-MBB was not significantly different from that of the 13-o-MBB regioisomers 13-(3-methylbenzyl) berberine bromide and 13-(4-methylbenzyl) berberine bromide, although the drug resistance inhibitory action of 13-o-MBB on the MexXY system is greater than that of these two regioisomers. This indicates that 13-o-MBB has greater inhibitory action against MexXY-dependent drug resistance than berberine and the other berberine derivatives we synthesized.

The deletion of *mexXY* from PAGU 1606 strain generated the PAGU^g^1659 strain. The addition of 13-o-MBB increased PAGU^g^1659 resistance towards norfloxacin, erythromycin and azithromycin 2-fold to 4-fold. Norfloxacin, erythromycin and azithromycin are substrates for MexCD-OprM and increased resistance towards norfloxacin, erythromycin and azithromycin may be due to the induction of MexCD-OprJ [24].

The addition of 13-o-MBB 256 µg/mL increased the efficacies of azithromycin and gentamicin to a Clinical and Laboratory Standards Institute (CLSI) breakpoint (amikacin is 64 µg/mL, gentamycin is 16 µg/mL) in a clinical strain of *P. aeruginosa* highly resistant to aminoglycosides. Amino acid residue Y613 within the loop of the drug binding pocket of MexY is directly involved in the recognition of aminoglycoside drugs, based on a decrease in sensitivity to aminoglycoside drugs upon mutation of Y613 have been reported [25]. Tobramycin and berberine have been reported to compete for Y613 on the docking simulations of tobramycin or berberine on MexY [16]. Furthermore, they claimed that the results of a combined berberine/tobramycin assay on different clinical isolates of *P. aeruginosa* were consistent with the in silico findings [16]. The results of our combination assay using berberine and 13-o-MBB with aminoglycosides are consistent with this report [16] and substantiate that the main mechanism of action of berberine and 13-o-MBB is competition for MexY inhibition. Another possible mechanism is suppression of MexY expression. However, Berberine decreased MexY mRNA only 0.8 to 0.9-fold have been reported [26]. Another reported that the MIC of amikacin and gentamicin was increased only up to 4-fold even in a strain *P. aeruginosa* that expresses 10–21 times more MexY mRNA than the PAO1 strain [12]. Our study of inhibited resistance by berberine showed that the gentamicin MIC for PAGU1606 was reduced 4-fold to 16-fold by berberine (Table 4), suggesting that the inhibition of MexY expression is not the main mechanism of action of berberine and 13-o-MBB.

13-o-MBB showed cytotoxicity against Caco-2 cells, a human epithelial colorectal adenocarcinoma cell line, at 30 µg/mL (data not shown). Thus, a concentration of 256 µg/mL 13-o-MBB could be toxic to human cells. There is thus a need to synthesize a compound that exhibits inhibitory action against MexXY system-dependent drug resistance at a lower concentration than 13-o-MBB and that is non-toxic to human cells.

## 4. Materials and Methods

### 4.1. Bacterial Strains and Growth Conditions

The bacterial strains used in this study are described in Table 7. Bacterial cells were grown in Luria (L) broth and on L agar (1.5%) under aerobic conditions at 37 °C, as previously described [27].

### 4.2. Antibiotic Susceptibility Assay

MICs were assessed in cation-adjusted Mueller–Hinton (MH) broth after about 18–22 h of incubation at 37 °C (for *P. aeruginosa*) or after about 20–24 h of incubation at 35 °C (for *A. xylosoxidans* and *B. cepacia*) using the two-fold serial micro-titer broth dilution method described previously [15]. The categorization as susceptible, intermediate, and resistant was performed according to the interpretive standards of the CLSI.

The FIC index was calculated as described previously [15]. The effects of the drugs were interpreted to be indicative of synergy when the index was ≤0.5.

### 4.3. Time-Killing Assay

We examined the bactericidal activity of gentamicin monotherapy or combination therapy with berberine or berberine derivatives towards PAGU^g^1929 and PAGU^g^1933. Each measurement was started by inoculating between 5 × 10^6^ to 2 × 10^7^ CFU/mL in cation-adjusted MH broth and incubating at 150 rpm at 37 °C on a shaker. Samples were withdrawn to measure the survival counts on MH agar plates at 0, 1, 2, 3 and 4 h. The MH agar plates were incubated at 37 °C for 16–18 h. The concentrations of drugs tested were gentamicin 2 μg/mL, berberine 256 μg/mL, and 13-o-MBB 64 μg/mL. The fraction surviving vs. the control for each sample was determined by taking the average CFU/mL values of the treated samples and dividing by the value for the same sample at 0 h. Each experiment was repeated at least three times, and a representative experiment is shown.

### 4.4. Synthesis

#### 4.4.1. General Synthesis Information

Melting points were measured on a Yanagimoto micro melting point hot-stage apparatus (MP-S3) and are reported as uncorrected values. ^1^H-NMR (TMS: δ: 0.00 ppm as an internal standard) and ^13^C-NMR (CDCl_3_: δ: 77.00 or DMSO-*d*_6_: 39.52 ppm as an internal standard) spectra were recorded on JEOL JNM-AL400 (400 MHz and 100 MHz) spectrometers in CDCl_3_ or DMSO-*d*_6_. Mass spectra were obtained on a JEOL JMP-DX300 instrument (70 eV, 300 mA). Chromatographic separations were accomplished using silica gel 60N (Kanto Chemical Co., Inc., Tokyo, Japan) or aluminum oxide 90 standardized (Merck KGaA., Inc., Darmstadt, Germany). Thin-layer chromatography (TLC) was performed using silica gel 60F254 and aluminum oxide 60F254 neutral (Merck KGaA, Inc., Darmstadt, Germany). All reagents were purchased from Wako Pure Chemical Industry, Osaka, Japan. Kanto Chemical Co., Inc., Tokyo, Japan. Tokyo Chemical Industry Co., Ltd., Tokyo, Japan. Kishida Chemical Co., Ltd., Osaka Japan and Sigma-Aldrich Co., LLC. St. Louis, MO, USA. Dihydroberberine was synthesized by the reduction of berberine according to the reported procedure [21].

#### 4.4.2. 13-Benzylberberine Derivatives; General Procedure

Each benzyl bromide (1.0 mmol) was added in a dropwise manner to a stirred solution of KI (310 mg, 1.86 mmol, 1.86 equiv) and dihydroberberine (337 mg, 1.0 mmol, 1 equiv) in CH_3_CN (40 mL), and the resulting mixture was held at reflux for 4 h. The reaction mixture was then filtered, and the filtrate was collected and evaporated to dryness in vacuo to give the crude residue. The residue was purified by column chromatography over neutral alumina using CHCl_3_/CH_3_OH (50:1 to 20:1) as eluent and recrystallization to give the final compounds 1–11. Compounds 1–10 were known compounds and their characterisation data were identical to those given in the literature. Their melting points (m.p.) were as follows: Compound 1; m.p. 198–200 °C [21], Compound 2; m.p. 179–180 °C [21], Compound 3; m.p. 235–240 °C [21], Compound 4; m.p. 214–216 °C [21], Compound 5; m.p. 210–211 °C [34], Compound 6; m.p. 218–220 °C [35], Compound 7; m.p. 216–220 °C [36], Compound 8; m.p. 222–225 °C [21], Compound 9; m.p. 204–207 °C [36] and Compound 10; m.p. 22–230 °C [21].

#### 4.4.3. Characterisation Data of 13-(2,6-Dichlorobenzyl)berberine Bromide (11)

Compound 11 is a yellow solid. Yield: 41%. ^1^H-NMR (DMSO-*d*_6_) δ: 9.95 (1H, s), 8.11 (1H, d, *J* = 9.3 Hz), 7.84 (1H, d, *J* = 9.3 Hz), 7.56 (1H, s), 7.37 (2H, d, *J* = 7.8 Hz), 7.23 (1H, t, *J* = 8.3 Hz), 7.15 (1H, s), 6.18 (2H, s), 5.16 (2H, s), 4.84 (2H, br), 4.09 (3H, s), 4.01 (3H, s), 3.08 (2H, br). ^13^C-NMR (DMSO-*d*_6_) δ: 150.0 (s), 149.3 (s), 146.5 (s), 144.2 (d), 144.1 (s), 138.2 (s), 134.9 (s), 134.7 (s), 133.6 (s), 131.9 (s), 131.3 (s), 129.5 (d), 129.2 (d), 125.8 (d), 121.0 (s), 120.5 (d), 120.4 (s), 110.9 (d), 108.1 (d), 102.0 (t), 62.0 (q), 56.9 (q), 56.7 (t), 32.9 (t), 27.4 (t). MS m/z: 494 (M–Br)^+^, 119, 85. m.p. 228–231 °C.

## 5. Conclusions

Eleven berberine derivatives were synthesized and tested for MexXY-dependent inhibition of gentamicin resistance using a *Pseudomonas aeruginosa* positive-MexXY strain and a negative-MexXY strain. 13-o-MBB showed the greatest inhibitory effect on MexXY-dependent gentamicin resistance. Regioisomers of 13-o-MBB exhibited no greater MexXY-dependent inhibition of gentamicin resistance than berberine. 13-o-MBB inhibited resistance to aminoglycosides 4-fold to 16-fold compared with berberine against the four tested *P. aeruginosa* clinical strains, and *Achromobacter xylosoxidans* and *Burkholderia cepacia*. These results indicate that 13-o-MBB inhibits the resistance to aminoglycosides in a MexXY-dependent manner more strongly than berberine. 13-o-MBB is thus a useful inhibitor of aminoglycoside drug resistance due to MexXY.

## Figures and Tables

**Figure 1 antibiotics-08-00212-f001:**
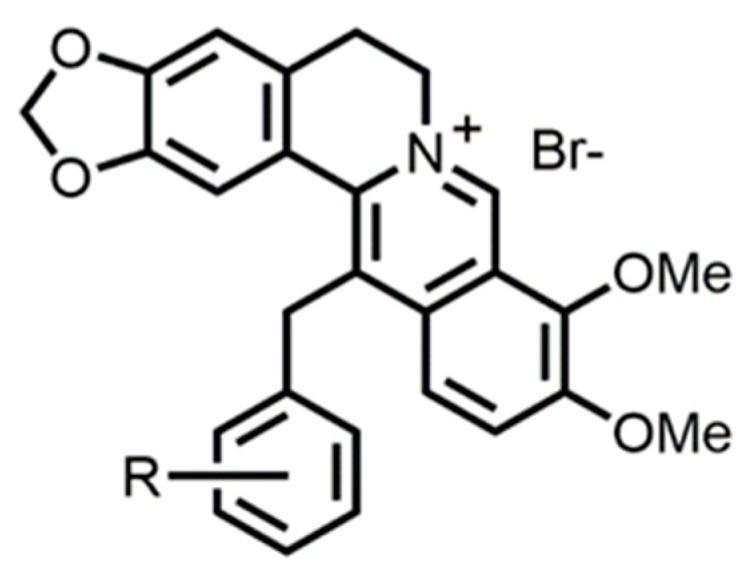
Structure of berberine derivatives.

**Figure 2 antibiotics-08-00212-f002:**
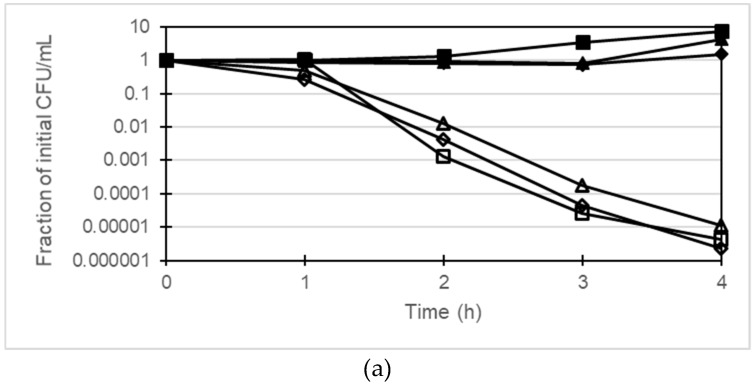
(**a**) Time-kill curves by the combination of gentamicin with berberine or 13-o-MBB against PAGU^g^1933, closed squares, control; closed triangles, berberine 256 µg/mL; closed diamonds, 13-o-MBB 64 µg/mL; open squares, gentamicin 2 µg/mL; open triangles, gentamicin 2 µg/mL with berberine 256 µg/mL; open diamonds, gentamicin 2 µg/mL with 13-o-MBB 64 µg/mL; (**b**) Time-kill curves by the combination of gentamicin with berberine or 13-o-MBB against PAGU^g^1929, closed squares, control; closed triangles, berberine 256 µg/mL; closed diamonds, 13-o-MBB 64 µg/mL; open squares, gentamicin 2 µg/mL; open triangles, gentamicin 2 µg/mL with berberine 256 µg/mL; open diamonds, gentamicin 2 µg/mL with 13-o-MBB 64 µg/mL.

**Table 1 antibiotics-08-00212-t001:** Antibacterial activities of berberine derivatives against PAGU^g^1931 and PAGU^g^1927.

Compound	-R	MIC of (µg/mL)	
		PAGU^g^1927	PAGU^g^1931
GM^2^	-	1024	8
Ber^3^	-	>512	>512
1	-H	256	128
2	*o*-Br	256	128
3	*p*-Br	128	64
4	*o*-F	512	256
5	*o*-Cl	256	128
6	*p*-Cl	256	128
7	*o*-CH_3_	512	256
8	*m*-CH_3_	256	128
9	*p*-CH_3_	256	128
10	*o*-NO_2_	512	256
11	2,6-Cl	128	64

Note: R, side chain of the benzyl group of 13-benzylberberine derivatives; GM, gentamicin; Ber, berberine.

**Table 2 antibiotics-08-00212-t002:** Increase in sensitivity to gentamicin by combination with berberine derivatives.

Concomitant Compound	-R	GM MIC with Berberine Derivative (µg/mL)											
		PAGU^g^1927						PAGU^g^1931					
		256 *	128	64	32	16	8	256	128	64	32	16	8
**Ber**	−	128	256	256	512	-	−	8	8	8	8	−	−
**1**	-H	−	−	32	64	128	−	−	−	8	8	8	−
**2**	*o*−Br	−	−	32	64	64	−	−	−	4	8	8	−
**3**	*p*−Br	−	−	−	128	256	512	−	−	−	4	8	8
**4**	*o*−F	−	32	64	128	−	−	−	4	8	8	−	−
**5**	*o*−Cl	−	−	32	64	128	−	−	−	4	8	8	−
**6**	*p*−Cl	−	−	256	256	256	−	−	−	4	8	8	−
**7**	*o*−CH_3_	−	16	32	64	−	−	−	4	8	8	−	−
**8**	*m*−CH_3_	−	−	256	256	512	−	−	−	4	8	8	−
**9**	*p*−CH_3_	−	−	256	256	512	−	−	−	4	8	8	−
**10**	*o*−NO_2_	−	128	128	256	−	−	−	8	8	8	−	−
**11**	2,6−Cl	−	−	−	128	256	512	−	−	−	4	8	8

Note: R, side chain of the benzyl group of 13−benzyl−berberine derivatives; GM, gentamicin; Ber, berberine, *; combined concentration (µg/mL).

**Table 3 antibiotics-08-00212-t003:** Increase in sensitivity to antibiotic resistance due to 13-o-MBB and its regioisomers.

Drug	MIC in the Presence of Berberine Derivative (µg/mL)									
	PAGU^g^1927					PAGU^g^1931				
	Ber	13-o-MBB	13−m−MBB	13−p−MBB	−	Ber	13-o-MBB	13−m−MBB	13−p−MBB	−
AMK	4	2	4	4	8	1	0.5	0.5	0.5	1
TOB	0.25	0.5	0.5	0.5	0.5	0.25	0.25	0.5	0.125	0.25
KM	128	64	256	256	512	64	32	32	64	64
SPCM	256	64	128	128	1024	64	8	8	32	64
NLFX	0.25	0.0625	1	1	1	0.015625	0.015625	0.015625	0.015625	0.015625
EM	128	64	512	256	512	16	16	16	8	16
CBPC	1	1	1	1	1	1	1	1	1	1
EtBr	512	128	256	256	256	64	8	8	16	64
Tc	2	1	8	8	16	0.125	0.125	0.125	0.125	0.25
Cp	4	2	8	8	8	2	2	2	1	2
AZM	64	64	256	256	512	8	8	8	4	8
CEF	1	0.25	0.125	2	8	0.0625	0.125	0.125	0.125	0.125

Note: Ber, combined berberine 256 µg/mL; 13-o-MBB, combined 13-o-MBB 128 µg/mL; 13−m−MBB, combined 13−m−MBB 64 µg/mL; 13−p−MBB, combined 13−p−MBB 64 µg/mL; AMK, amikacin; TOB, tobramycin; KM, kanamycin; SPCM, spectinomycin; NLFX, norfloxacin; EM, erythromycin; CBPC, carbenicillin; EtBr, ethidium bromide; Tc, tetracycline; Cp, chloramphenicol; AZM, azithromycin; CEF, cefepime.

**Table 4 antibiotics-08-00212-t004:** Inhibited resistance to aminoglycoside-based drugs by 13-o-MBB in PAGU 1606

Drug	MIC (µg/mL)													
	PAGU 1606							PAGU^g^1659						
	-	Ber (256) ^1^	Ber (128)	Ber (64)	13-o-MBB (256)	13-o-MBB (128)	13-o-MBB (64)	-	Ber (256)	Ber (128)	Ber (64)	13-o-MBB (256)	13-o-MBB (128)	13-o-MBB (64)
AMK	256	64	128	128	16	32	32	16	8	8	8	8	8	8
TOB	256	64	128	128	16	32	32	8	8	8	8	8	8	8
KM	>2048	1024	1024	2048	256	512	1024	256	256	256	256	256	256	256
GM	64	4	8	16	2	4	4	0.5	0.25	0.25	0.5	0.25	0.25	0.5
SPCM	>2048	>2048	>2048	>2048	2048	>2048	>2048	2048	2048	2048	2048	1024	2048	2048
NLFX	256	256	256	256	256	256	256	64	256	256	256	128	128	128
CPFX	64	64	64	64	32	32	32	64	64	64	64	32	32	32
EM	256	128	256	256	128	256	256	128	256	256	256	256	256	256
CBPC	>512	>512	>512	>512	>512	>512	>512	>512	>512	>512	>512	>512	>512	>512
EtBr	>512	>512	>512	>512	>512	>512	>512	>512	>512	>512	>512	>512	>512	>512
Tc	32	16	16	16	16	16	16	16	16	16	16	16	16	16
Cp	128	64	128	128	64	64	64	128	128	128	128	128	128	128
AZM	256	64	64	128	64	64	128	32	256	256	256	128	128	128
CEF	512	512	512	512	512	512	512	512	512	512	512	512	512	512

Note: ^1^, values in parentheses are combined concentrations (µg/mL); Ber, berberine; AMK, amikacin; TOB, tobramycin; KM, kanamycin; GM, gentamicin; SPCM, spectinomycin; NLFX, norfloxacin; CPFX, ciprofloxacin; EM, erythromycin; CBPC, carbenicillin; EtBr, ethidium bromide; Tc, tetracycline; Cp, chloramphenicol; AZM, azithromycin; CEF, cefepime.

**Table 5 antibiotics-08-00212-t005:** Inhibition by 13-o-MBB of aminoglycoside resistance in *P. aeruginosa* clinical strains.

Strain	MIC of Aminoglycoside (µg/mL)														
	AMK			GM			TOB			KM			SPEC		
	−	Ber	13-o-MBB	−	Ber	13-o-MBB	−	Ber	13-o-MBB	−	Ber	13-o-MBB	−	Ber	13-o-MBB
PAGU 0974	4	1	0.5	4	0.5	0.25	0.5	0.125	0.125	128	32	32	512	128	32
PAGU^g^ 0975	1	0.5	0.5	0.25	0.125	0.25	0.25	0.25	0.125	64	32	32	32	32	32
PAGU 1498	32	8	1	1024	128	8	256	32	8	>2048	512	256	512	128	32
PAGU^g^1565	2	1	1	8	8	8	8	8	8	512	256	256	32	32	32
PAGU 1569	256	64	32	256	32	8	16	8	4	>2048	>2048	1024	512	256	128
PAGU^g^1627	32	32	32	8	8	8	8	8	4	1024	512	1024	128	128	128
*PAGU 0013	128	32	4	128	32	4	64	8	1	64	8	2	1024	128	16
PAGU 0002	>2048	256	16	>2048	32	4	512	16	4	>2048	2048	256	>2048	512	64

**Table 6 antibiotics-08-00212-t006:** Antibacterial activities of berberine derivatives against *P. aeruginosa.*

**Strain**	**MIC (µg/mL) for AMK in the Presence of:**		**MIC (µg/mL) for 13-o-MBB in the Presence of:**		**FIC**	**Mode of Interaction**
	**−**	**13-o-MBB**	**−**	**AMK**		
PAGU^g^1931	1	1	256	256	2.0	Indifferent
PAGU^g^1927	8	2	512	128	0.5	Synergy
PAGU^g^1659	16	8	>512	>512	>1.5	Indifferent
PAGU 1606	256	16	>512	64	<0.5	Synergy
**Strain**	**MIC (µg/mL) for AMK in the Presence of:**		**MIC (µg/mL) for Berberne in the Presence of:**		**FIC**	**Mode of Interaction**
	**−**	**Berberine**	**−**	**AMK**		
PAGU^g^1931	1	1	>512	>512	>2.0	Indifferent
PAGU^g^1927	8	4	>512	512	<1.0	Synergy or Addition
PAGU^g^1659	16	8	>512	>512	1.5	Indifferent
PAGU 1606	256	64	>512	512	<0.75	Synergy or Addition
**Strain**	**MIC (µg/mL) for GM in the Presence of:**		**MIC (µg/mL) for 13-o-MBB in the Presence of:**		**FIC**	**Mode of Interaction**
	**−**	**13-o-MBB**	**−**	**GM**		
PAGU^g^1931	8	8	256	256	2.0	Indifferent
PAGU^g^1927	1024	32	512	4	0.04	Synergy
PAGU^g^1659	0.5	0.5	>512	>512	>1.0	Indifferent
PAGU 1606	64	2	>512	8	<0.5	Synergy
**Strain**	**MIC (µg/mL) for GM in the Presence of:**		**MIC(µg/mL) for Berberine in the Presence of:**		**FIC**	**Mode of Interaction**
	**−**	**Berberine**	**−**	**GM**		
PAGU^g^1931	8	8	>512	>512	>1.0	Indifferent
PAGU^g^1927	1024	128	>512	8	<0.5	Synergy
PAGU^g^1659	0.5	0.5	>512	>512	>1.0	Indifferent
PAGU 1606	64	8	>512	256	<0.5	Synergy

Note: GM, gentamicin; AMK, amikacin; FIC, fractional inhibitory concentration index.

**Table 7 antibiotics-08-00212-t007:** Bacterial strains and gene properties.

Strain Name	Relevant Characteristics	Reference
*Pseudomonas aeruginosa*		
PAGU 0974	PAO1 (K. Poole Lab), wild type	[28]
PAGU^g^0975	PAGU 0974Δ*mexXY*	[29]
PAGU 1498	PA7 Non-respiratory clinical isolate	[8]
PAGU^g^1565	PA7Δ*mexXY-oprA*	[8]
PAGU 1569	K2162 Pan-aminoglycoside-resistant clinical isolate	[30]
PAGU 1606	NCGM2. S1 Multidrug-resistant clinical isolate	[31]
PAGU^g^1627	K2162Δ*mexXY*	[30]
PAGU^g^1659	PAGU 1606Δ*mexXY*	[8]
PAGU^g^1927	YM34 Δ*mexZ*, *mexVW:: gfp-aacC1*	[15]
PAGU^g^1929	YM34 Δ*mexZ*, *mexVW*	[15]
PAGU^g^1931	PAGU^g^1927::Δ*mexXY*	[15]
PAGU^g^1933	PAGU^g^1929::Δ*mexXY*	[15]
Others		
PAGU 0002	ATCC 27061 *Achromobacter xylosoxidans* subsp. *xylosoxidans*	[32]
PAGU 0013	ATCC 25416 *Burkholderia cepacia*	[33]

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
