# Peer review of "13-(2-Methylbenzyl) Berberine Is a More Potent Inhibitor of MexXY-Dependent Aminoglycoside Resistance than Berberine"

_antibiotics, 2019, doi:10.3390/antibiotics8040212_

Round 1

Reviewer 1 Report

Abstract is not well organized.

The introduction is too short. other antimicrobial materials, e.g. quaternary ammonium compounds, should be iintroduced I recommend (Polymeric and inorganic nanoscopical antimicrobial fillers in dentistry Acta biomaterialia 2019;

Progresses in conductive polyaniline-based nanocomposites for biomedical applications: A review, Journal of Medicinal Chemistry 2019)

Conclusion should be added

Fig 1 has low quality. Please replace it.

Fig. 2 must be replaced. It is recommended to use a better software for depicting the graphs.

Author Response

Response to Reviewer 1 Comments

We appreciate you taking the time to offer us your comments and insights related to the paper. We found your feedback very constructive.

Point 1: Abstract is not well organized.

Response 1: We added the following to the end of the Abstract (p. 1, lines 25-27): " Our findings indicate that 13-o-MBB inhibits MexXY-dependent aminoglycoside drug resistance more strongly than berberine and that 13-o-MBB is a useful inhibitor of aminoglycoside drug resistance due to MexXY."

Point 2: The introduction is too short. other antimicrobial materials, e.g. quaternary ammonium compounds, should be iintroduced I recommend (Polymeric and inorganic nanoscopical antimicrobial fillers in dentistry Acta biomaterialia 2019;

Response 2: Thank you for your suggestions and references. We have significantly modified the Introduction.

Point 3: Conclusion should be added

Response 3: We added a Conclusion section as follows (p 11, Line 273-282): " Eleven berberine derivatives were synthesized and tested for MexXY-dependent inhibition of gentamicin resistance using a Pseudomonas aeruginosa positive-MexXY strain and a negative-MexXY strain. 13-o-MBB showed the greatest inhibitory effect on MexXY-dependent gentamicin resistance. Regioisomers of 13-o-MBB exhibited no greater MexXY-dependent inhibition of gentamicin resistance than berberine. 13-o-MBB inhibited resistance to aminoglycosides 4- to 16-fold compared with berberine against the four tested P. aeruginosa clinical strains, and Achromobacter xylosoxidans and Burkholderia cepacia. These results indicate that 13-o-MBB inhibits the resistance to aminoglycosides in a MexXY-dependent manner more strongly than berberine. 13-o-MBB is thus a useful inhibitor of aminoglycoside drug resistance due to MexXY.."

Thank you once again for your valuable comments and suggestions. We hope you find our revised manuscript satisfactory.

Reviewer 2 Report

This manuscript addresses the important issue of MexXY-dependent aminoglycoside resistance in Pseudomonas aeruginosa and the data presented generally support the conclusions. However there are several confusing statements in the manuscript, which reduce its clarity. Perhaps more importantly, there are no experiments and there is no discussion directed at the precise mechanism through which berberine and 13-o-MBB inhibit MexXY.

1. Lines 100-101 state, "However, the MICs of norfloxacin, erythromycin and azithromycin were increased towards PAGU81659 and PAGU81606, which are pump deficient strains." Isn't only PAGU81659 a pump deficient strain? Also, based on Table 4, the MICs for those three drugs were only increased in the PAGU81659 strain, not in the PAGU81606 strain. Please clarify this statement.

2. Lines 150-151 state, "...in the MexXY-positive P. aeruginosa strain PAGU81933..." But based on Figure 2A, the time-kill curve of PAGU81933 is not affected by the addition of berberine or 13-o-MBB...Isn't PAGU81933 the MexXY deficient strain and PAGU81929 the MexXY-positive strain? Please clarify this statement.

3. Lines 158-159 state, in referring to the PAGU81659 strain, "The addition of 13-o-MBB increased the efficacy of norfloxacin, erythromycin and azithromycin 2- to 4-fold." However, this is not the case, based on Table 4. Also, why would this result be expected, if PAGU81659 is MexXY deficient? Please clarify this statement. 

In addition to my concern about confusing statements in the manuscript such as the ones above, I would request that the mechanism whereby your compounds reduce MexXY-dependent drug resistance be more directly addressed. Do berberine and 13-o-MBB inhibit efflux of aminogycosides from bacterial cells by inhibiting MexXY? Are your compounds substrates for this transporter? Are they competitive inhibitors? Could accumulation and efflux assays with labeled aminoglycosides be performed to address this question? Lines 134-135 state, "...showing that the combination of amikacin and gentamicin in the presence of MexXY increases the accumulation of 13-o-MBB and berberine in the cell," implying that the drugs compete for the same transporter. But can you show that 13-o-MBB concentrations are increased in the bacterial cells by performing accumulation assays with labeled 13-o-MBB? Is it possible that berberine and 13-o-MBB reduce expression of MexXY rather than inhibit drug efflux? Can you show that expression of the transporter is stable?

I have a few additional concerns, which are more minor.

Lines 162-164 mention data involving a clinical strain of P. aeruginosa. Is that statement meant to summarize data presented in this paper or in the literature? If in the literature, please provide the citation.

The widths of the table columns must be adjusted so that the column titles are not broken up, in Tables 3, 4 and especially 5.

In Line 80, "t" should be "to."

Author Response

Response to Reviewer 2 Comments

Thank you for your detailed comments and suggestions, which we found most helpful as we revised our manuscript.

Point 1: Lines 100-101 state, "However, the MICs of norfloxacin, erythromycin and azithromycin were increased towards PAGU81659 and PAGU81606, which are pump deficient strains." Isn't only PAGU81659 a pump deficient strain? Also, based on Table 4, the MICs for those three drugs were only increased in the PAGU81659 strain, not in the PAGU81606 strain. Please clarify this statement.

Response 1: The reviewer's comment is correct. To clarify, we have added the following text to the Results (p. 4, lines 126-128): " However, the MICs of norfloxacin, erythromycin and azithromycin were increased towards PAGUg1659, a pump-deficient strain. "

The MexXY-dependent resistance-inhibiting action of berberine and its derivatives reduces MexXY resistance 2- to 4-fold, but due to induction of MexCD by membrane damage, the MexXY inhibitory activity of these drugs is not observed.

Point 2: Lines 150-151 state, "...in the MexXY-positive P. aeruginosa strain PAGU81933..." But based on Figure 2A, the time-kill curve of PAGU81933 is not affected by the addition of berberine or 13-o-MBB...Isn't PAGU81933 the MexXY deficient strain and PAGU81929 the MexXY-positive strain? Please clarify this statement.

Response 2: The reviewer's comment is correct. To clarify, we added the following text to the Discussion (p. 8, lines 178-180): " The addition of 128 µg/mL 13-o-MBB increased the sensitivity to all aminoglycosides by 2- to 8-fold in comparison with 256 µg/mL berberine in the MexXY-positive P. aeruginosa strain PAGUg1927"

Point 3: Lines 158-159 state, in referring to the PAGU81659 strain, "The addition of 13-o-MBB increased the efficacy of norfloxacin, erythromycin and azithromycin 2- to 4-fold." However, this is not the case, based on Table 4. Also, why would this result be expected, if PAGU81659 is MexXY deficient? Please clarify this statement.

Response 3: The reviewer's comment is correct. To clarify, we added the following text to the Discussion (p. 9, Lines 186-188): " The addition of 13-o-MBB increased PAGUg1659 resistance towards norfloxacin, erythromycin and azithromycin 2- to 4-fold." We believe that resistance is increased by the induction of MexCD-OprM by membrane damage due to berberine and 13-o-MBB.

Point 4: In addition to my concern about confusing statements in the manuscript such as the ones above, I would request that the mechanism whereby your compounds reduce MexXY-dependent drug resistance be more directly addressed. Do berberine and 13-o-MBB inhibit efflux of aminogycosides from bacterial cells by inhibiting MexXY? Are your compounds substrates for this transporter? Are they competitive inhibitors? Could accumulation and efflux assays with labeled aminoglycosides be performed to address this question? Lines 134-135 state, "...showing that the combination of amikacin and gentamicin in the presence of MexXY increases the accumulation of 13-o-MBB and berberine in the cell," implying that the drugs compete for the same transporter. But can you show that 13-o-MBB concentrations are increased in the bacterial cells by performing accumulation assays with labeled 13-o-MBB? Is it possible that berberine and 13-o-MBB reduce expression of MexXY rather than inhibit drug efflux? Can you show that expression of the transporter is stable?

Response 4: We have added the following to the Discussion to clarify these points out (p. 9, Line 193-208): " Lau et al. reported that the loop of the drug binding pocket of MexY includes amino acid residue Y613 and that this residue is directly involved in the recognition of aminoglycoside drugs, based on a decrease in sensitivity to aminoglycoside drugs upon mutation of Y613 [28]. Laudadio et al. performed docking simulations of tobramycin or berberine with MexY and observed that tobramycin and berberine compete for Y613 [19]. Furthermore, they claimed that the results of a combined berberine/tobramycin assay on different clinical isolates of P. aeruginosa were consistent with the in silico findings [19]. The results of our combination assay using berberine and 13-o-MBB with aminoglycosides are consistent with the report of Laudadio et al. and substantiate that that the main mechanism of action of berberine and 13-o-MBB is competition for MexY inhibition. Another possible mechanism is suppression of MexY expression. However, Su et al. reported that berberine decreased MexY mRNA only 0.8-0.9-fold [29]. Islam et al. reported that the MIC of amikacin and gentamicin was increased only up to 4-fold even in a strain P. aeruginosa that expresses 10-21 times more MexY mRNA than the PAO1 strain [15]. Our study of inhibited resistance by berberine showed that the gentamicin MIC for PAGU 1606 was reduced 4- to 16-fold by berberine (Table 4), suggesting that the inhibition of MexY expression is not the main mechanism of action of berberine and 13-o-MBB."

We attempted to perform an accumulation assay using 13-o-MBB, but were not successful.

Again, we appreciate your insightful comments and hope we have satisfactorily addressed all your concerns regarding our manuscript. Thank you for taking the time to help us improve the paper.

Round 2

Reviewer 1 Report

Corresponding author is not have *. please specify him/her.

Line 186: This is not well written and only is X et al. reported that...., Please revise this section.

Fig. 2 should be re-depicted. Please use a better graph. Time [h] not hr is correct. Also please use Arial font in the graph.

Some of the newly added references are old. They should be replaced. They did not use the recommended reference as well.

It has many self citation, e.g Nishino 3 times, Kawamura 6 times. Some of the ref are too old, e.g. Poole, K,; Gotoh, N.; Tsujimoto, H.; Zhao, Q.; Wada, A.; Yamasaki, T.; Neshat, S.; Yamagishi, J.; Li, X. Z.;
Nishino, T. Overexpression of the mexC-mexD-oprJ efflux operon in nfxB-type multidrug-resistant strains
of Pseudomonas aeruginosa. Mol Microbiol. 1996. 21(4):713-24.

Author Response

Response to Reviewer 1 Comments

Thank you for your suggestion. We agree with you and have incorporated this suggestion throughout our paper.

Point 1:Corresponding author is not have *. please specify him/her.

Response 1:Thank you for pointing it out. We added the following text to Author information.

p. 1, Line 6: Yoshiaki Kawamura 1, *

p. 1, Line 13: *Author to whom correspondence should be addressed.

Point 2:Line 186: This is not well written and only is X et al. reported that...., Please revise this section.

Response 2: Thank you for your advice. We modified our text in the Discussion (p. 8, Line 187-199):

“Amino acid residue Y613 within the loop of the drug binding pocket of MexY is directly involved in the recognition of aminoglycoside drugs, based on a decrease in sensitivity to aminoglycoside drugs upon mutation of Y613 have been reported [28]. Tobramycin and berberine have been reported to compete for Y613 on the docking simulations of tobramycin or berberine on MexY [19].Furthermore, they claimed that the results of a combined berberine/tobramycin assay on different clinical isolates of P. aeruginosa were consistent with the in silico findings [19]. The results of our combination assay using berberine and 13-o-MBB with aminoglycosides are consistent with this report [19] and substantiate that the main mechanism of action of berberine and 13-o-MBB is competition for MexY inhibition. Another possible mechanism is suppression of MexY expression. However, Berberine decreased MexY mRNA only 0.8-0.9-fold have been reported [29]. Another reported that the MIC of amikacin and gentamicin was increased only up to 4-fold even in a strain P. aeruginosa that expresses 10-21 times more MexY mRNA than the PAO1 strain [15]. “

Point 3: Fig. 2 should be re-depicted. Please use a better graph. Time [h] not hr is correct. Also please use Arial font in the graph.

Response 3: Thank you for pointing it out. We changed Fig. 2 (using Arial font and Time [h]).

Point 4: Some of the newly added references are old. They should be replaced. They did not use the recommended reference as well.

It has many self citation, e.g Nishino 3 times, Kawamura 6 times. Some of the ref are too old, e.g. Poole, K,; Gotoh, N.; Tsujimoto, H.; Zhao, Q.; Wada, A.; Yamasaki, T.; Neshat, S.; Yamagishi, J.; Li, X. Z.;Nishino, T. Overexpression of the mexC-mexD-oprJ efflux operon in nfxB-type multidrug-resistant strains of Pseudomonas aeruginosa. Mol Microbiol. 1996. 21(4):713-24.

Response 4: We added the following text to the Introduction(p. 2, line 61-62)

“In addition, quaternary ammonium compounds inhibit the biofilm formation in P. aeruginosa and C. albicans have been reported [22].”

Reference [22] mean your suggested report entitled as “Polymeric and inorganic nanoscopical antimicrobial fillers in dentistry.”

And we changed or deleted some reference as much as possible according to your suggestion.

Poole, K.; Krebes, K.; McNally, C.; Neshat, S. Multiple antibiotic resistance in Pseudomonas aeruginosa: evidence for involvement of an efflux operon. J Bacteriol. 1993. 175(22): 7363–7372.( Ref 4)

-> Change to: Tsutsumi, K.; Yonehara, R.; Ishizaka-Ikeda, E.; Miyazaki, N.; Maeda, S.; Iwasaki, K.; Nakagawa, A.; Yamashita, E. Structures of the wild-type MexAB-OprM tripartite pump reveal its complex formation and drug efflux mechanism. Nat Commun. 2019. 10:1520. (New Ref [4])

We replaced to New report.

Poole, K.; Gotoh, N.; Tsujimoto, H.; Zhao, Q.; Wada, A.; Yamasaki, T.; Neshat, S.; Yamagishi, J.; Li, X. Z.; Nishino, T. Overexpression of the mexC-mexD-oprJ efflux operon in nfxB-type multidrug-resistant strains of Pseudomonas aeruginosa. Mol Microbiol. 1996. 21(4):713-24. (Ref [5])

-> Change to: Alcalde-Rico, M.; Olivares-Pacheco, J.; Alvarez-Ortega, C.; Cámara, M.; Martínez, JL. Role of the Multidrug Resistance Efflux Pump MexCD-OprJ in the Pseudomonas aeruginosa Quorum Sensing Response. Front Microbiol. 2018. 9:2752. (New Ref [5])

We replaced to New report.

Köhler, T.; Michéa-Hamzehpour, M.; Henze, U.; Gotoh, N.; Curty, L. K.; Pechère, J. C. Characterization of MexE-MexF-OprN, a positively regulated multidrug efflux system of Pseudomonas aeruginosa. Mol Microbiol. 1997. 23(2):345-54. (Ref [6])

-> Change to: Juarez, P.; Broutin, I.; Bordi, C.; Plésiat, P.; Llanes, C. Constitutive Activation of MexT by Amino Acid Substitutions Results in MexEF-OprN Overproduction in Clinical Isolates of Pseudomonas aeruginosa. Antimicrob Agents Chemother. 2018. 62(5):e02445-17.(New Ref [6])

We replaced to New report.

Mine, T.; Morita, Y.; Kataoka, A.; Mizushima, T.; Tsuchiya, T. Expression in Escherichia coli of a new multidrug efflux pump, MexXY, from Pseudomonas aeruginosa. Antimicrob Agents Chemother. 1997. 43(2):415-7.(Ref [7])

-> Deleted

Reason: New Ref No [12] can complement Ref [7].

Westbrock-Wadman, S.; Sherman, D.R.; Hickey, M. J.; Coulter, S. N.; Zhu, Y. Q.; Warrener, P.; Nguyen, L. Y.; Shawar, R. M.; Folger, K. R.; Stover, C. K. Characterization of a Pseudomonas aeruginosa efflux pump contributing to aminoglycoside impermeability. Antimicrob Agents Chemother. 1999. 43(12):2975-83. (Ref [9]) Aires, J. R.; Köhler, T.; Nikaido, H.; Plésiat, P. Involvement of an active efflux system in the natural resistance of Pseudomonas aeruginosa to aminoglycosides. Antimicrob Agents Chemother. 1999. 43(11):2624-8. (Ref [10])

-> Deleted.

Reason: The reports of New Ref No [9] and [10] can complement above two report information.

Tomkiewicz, D.; Casadei, G.; Larkins-Ford, J.; Moy, TI.; Garner, J.; Bremner, JB.; Ausubel, FM.; Lewis, K.; Kelso, MJ. Berberine-INF55 (5-nitro-2-phenylindole) hybrid antimicrobials: effects of varying the relative orientation of the berberine and INF55 components. Antimicrob Agents Chemother. 2010. 54(8):3219-24. (Ref [23])

-> Change to: Dolla, NK.; Chen, C.; Larkins-Ford, J.; Rajamuthiah, R.; Jagadeesan, S.; Conery, AL.; Ausubel, FM.; Mylonakis, E.; Bremner, JB.; Lewis, K.; Kelso, MJ. On the Mechanism of Berberine-INF55 (5-Nitro-2-phenylindole) Hybrid Antibacterials. Aust J Chem. 2015. 67:1471-80. (New Ref No [20])

Reason: We found new report.

We worked hard to be responsive to them. Thank you for taking the time and energy to help us improve the paper.

Reviewer 2 Report

Thank you for addressing all of my suggestions.

For Point #2, should it be PAGUg1929 or PAGUg1927? The Figure 2 legend mentions PAGUg1929, but the text mentions PAGUg1927.

The addition to the discussion section greatly improves the manuscript. 

Please correct the typo (that that) in Line 193.

Author Response

Response to Reviewer 2 Comments

We are grateful for the time and energy you expended on our behalf.

Point 1: (1) For Point #2, should it be PAGUg1929 or PAGUg1927? The Figure 2 legend mentions PAGUg1929, but the text mentions PAGUg1927. (2) The addition to the discussion section greatly improves the manuscript. (3) Please correct the typo (that that) in Line 193.

Response 1: (1) I’m so sorry for the confusion. The Figure 2 legend mentions PAGUg1929 was actually corrected and we explained the Time-killing assay result in the Line 161-168. PAGUg1927 mentioned in the Line 172-173 explain the result of Table 2 and 3. To clean the relationship, we added (Table 2 and 3) in the Line 173.

(2) Thank you so much.

(3) We corrected the typo (that that -> that) in Line 195.

Again, we appreciate all of your insightful comments.

Round 3

Reviewer 1 Report

acceptable